# Exposing Hallucinations to Suppress Them: VLMs Representation Editing with Generative Anchors

## Abstract

Vision language models (VLMs) have achieved remarkable success across diverse vision-language tasks, yet they remain highly susceptible to hallucinations, producing content that is fluent but inconsistent with visual evidence. Such hallucinations, spanning objects, attributes, and relations, persist even in larger models, while existing mitigation approaches often require additional fine-tuning, handcrafted priors, or trade-offs that compromise informativeness and scalability. To address this limitation, we propose a training-free, self-supervised method for hallucination mitigation. Our approach introduces a novel hallucination amplification mechanism: a caption is projected into the visual space via a text-to-image model to reveal implicit hallucination signals, serving as a negative anchor, while the original image provides a positive anchor. Leveraging these dual anchors, we edit decoder hidden states by pulling representations toward faithful semantics and pushing them away from hallucination directions. This correction requires no human priors or additional training costs, ensuring both effectiveness and efficiency. Extensive experiments across multiple benchmarks show that our method significantly reduces hallucinations at the object, attribute, and relation levels while largely preserving recall and caption richness, e.g., achieving a hallucination reduction by over 5% using LLaVA-v1.5-7B on CHAIR. Furthermore, results on diverse architectures, including LLaVA-NEXT-7B, Cambrian-8B, and InstructBLIP-7B, validate strong cross-architecture generalization. More importantly, when applied to hallucination-free captions, our method introduces almost no side effects, underscoring its robustness and practical plug-and-play applicability. The implementation will be publicly available.

## 1 Introduction

Vision Language Models (VLMs) (OpenAI, 2023b; Zhu & et al., 2025; Team et al., 2025a; Bai & et al., 2025; Liu et al., 2023; Team et al., 2025b; Lu et al., 2025) have achieved remarkable progress on diverse vision-language tasks, including image caption (Ge et al., 2024; Wang et al., 2023), visual question answering (Lee et al., 2024; Lin et al., 2025), and cross-modal retrieval (Bai et al., 2025b; Yang et al., 2024). Despite the progress, VLMs remain vulnerable to hallucinations, i.e., generating content that is fluent and plausible but inconsistent with the visual semantics. These hallucinations can be generally categorized into three types: object-level, attribute-level, and relation-level (Bai et al., 2025a). For instance, models may often mention non-existent objects, assign incorrect attributes, or describe spurious relations between objects.

Recent studies have shown that hallucinations persist even in larger and more advanced models (Rohrbach et al., 2019; Li et al., 2023b;a; Jiang et al., 2025), suggesting that scaling alone is insufficient. Existing efforts to mitigate hallucinations in VLMs often involve external interventions (e.g., object detector) (Yin et al., 2024), additional fine-tuning (Zhang et al., 2024; Zhou et al., 2023; Hu et al., 2024), or latent editing and decoding adjustments (Jiang et al., 2025; Liang et al., 2024). However, these approaches face limitations: they rely on hand-crafted priors, fine-tuning demands additional data and computational overheads, and many latent editing-based methods often struggle with subtle hallucinations and may inadvertently suppress useful semantics. Therefore, this raises a

Figure 1: Illustration of our proposed method on LLaVA-v1.5-7b. The reconstructed image from the regular model's caption contains clearer hallucinated objects (circled in red), with the corresponding hallucinated terms highlighted in **bold red** with the caption. In contrast, our method not only effectively removes hallucinations but also preserves the information richness of the caption, with more accurate and specific descriptions.

critical question: *Can we design a training-free and self-supervised method that requires no extra supervision to mitigate hallucination without sacrificing informativeness?*

To address this, we propose a training-free method that identifies correction directions to edit hidden states in the decoder layers in a self-supervised manner, after which the model produces captions that remain informative while being more faithfully grounded in visual evidence. Specifically, since hallucinations are typically subtle and even undetectable based solely on text modality, we find that reconstructing an image from a potentially hallucinated caption via a Text-to-Image (T2I) model (Ho et al., 2020) can expose implicit hallucination information, serving as one visual anchor during inference. Meanwhile, the original image provides the second visual anchor that guides the representation toward the ground truth semantics. Based on these dual anchors, we manipulate the latent representation by pulling image token embeddings toward the clean semantics of the original image and pushing them away from the hallucination direction derived from the reconstruction. In this way, our method establishes a fully self-supervised correction mechanism, requiring no human priors or additional training, while maintaining informativeness and robustness. More importantly, when applied to hallucination-free captions, our method introduces almost no side effects, enabling seamless integration into existing VLMs to reduce potential hallucinations without first detecting hallucinations. Experiment results across various benchmarks and models demonstrate that our approach can effectively reduce hallucinations at the object, attribute, and relation levels while largely preserving recall and caption richness. Moreover, results across different VLM architectures, e.g., LLaVA-v1.5-7B (Liu et al., 2023), LLaVA-NEXT-7B (Liu et al., 2024b), etc., demonstrate strong cross-architecture generalization and plug-and-play applicability.

The contributions of this work can be summarized as follows: **1)** We propose a training-free, self-supervised method to mitigate hallucinations in VLMs. By deriving supervision directly from the model's own outputs, our approach works in an entirely end-to-end and plug-and-play fashion. **2)** We introduce a novel hallucination amplification mechanism that projects caption semantics into

the visual space using a T2I model. This makes otherwise implicit hallucinations perceptible and provides a lightweight way to construct reliable supervisory signals. **3)** Our method jointly anchors semantics from the original images and suppresses hallucination directions from the reconstructed image. This dual steering guidance removes only hallucination components while preserving genuine semantics, striking a balance between faithfulness and informativeness. **4)** Experiment results show that our method outperforms existing approaches by a large margin. Meanwhile, our approach achieves the optimal trade-off between hallucination reduction and information richness, establishing a strong baseline for future research on hallucination reduction.

## 2 RELATED WORK

**Vision Large Language Models.** The emergence of VLMs (Team et al., 2025a; Zhu & et al., 2025) marks a significant advancement in Visual Question Answering (VQA), image captioning, and so on, extending the capabilities of traditional Large Language Models (LLMs) to process and reason across diverse modalities. VLMs fuse visual encoders, visual projectors, and LLMs, which could leverage visual components to let VLMs understand and reason the information mixed with image and text. For instance, GPT-4v (OpenAI, 2023b) builds upon GPT-4 (OpenAI et al., 2024), Qwen2.5-VL (Bai & et al., 2025) is based on Qwen2.5-LM (Qwen et al., 2025), and LLaVA (Liu et al., 2023) incorporates Vicuna (Chiang et al., 2023). Most VLMs follow a two-stage training paradigm, consisting of pre-training and post-training. The pre-training stage exposes the model to large-scale image-text data to learn general visual knowledge. Post-training then applies refined techniques, such as Supervised Fine-Tuning (SFT) (Dai et al., 2023) and reinforcement learning (e.g., RLHF (Ouyang et al., 2022; Schulman et al., 2017; Shao et al., 2024)), to improve downstream performance and better align with human preferences.

**Mitigating Hallucinations in VLMs.** Efforts to mitigate hallucinations in VLM can be categorized into two main directions: training-related and inference-related work (Bai et al., 2025a). Training-related strategies primarily involve auxiliary supervision, which uses visual signals (Chen et al., 2023) or leveraging contrastive learning (Sarkar et al., 2025), and reinforcement learning from human feedback (Ben-Kish et al., 2024; Sun et al., 2023; Yu et al., 2024) to obtain more reliable and trustworthy model outputs. Inference-related approaches offer lightweight and efficient alternatives that do not require retraining the model. Many of them put efforts into the decoding strategy, such as Contrastive Decoding VCD (Leng et al., 2023), ICD (Wang et al., 2024), and Guided Decoding DeCo (Wang et al., 2025). Some studies also explore modifying intermediate representations within the language model component of VLMs, which not only helps mitigate hallucinations but also offers a good interpretable insight (Jiang et al., 2025; Liu et al., 2024c). A great body of work has shown that these training-free or post-hoc revision methods can achieve or even outperform training-related methods (Ge et al., 2024), making them appealing for practical applications.

**Text-Image Generative Model Efforts in VLMs.** Text-to-Image (T2I) generative models have been increasingly integrated into the development and evaluation of VLMs, typically for two major purposes. First, T2I models are used for benchmark construction, where generated images are based on corresponding captions that include diverse open-vocabulary objects using a T2I diffusion model (Ben-Kish et al., 2024). Second, T2I models play a crucial role in the construction of training datasets. V-DPO (Xie et al., 2024) devises a vision-guided direct preference optimization with a synthetic dataset containing both response-contrast and image-contrast preference pairs. ES-REAL (Kim et al., 2024) proposes to utilize a T2I model to semantically reconstruct an image from the generated caption. After that, the semantic misalignment between the two images can serve as feedback to optimize the model. In addition, ConVis Park et al. (2024) also explore applying reconstructed image from T2I models during the model's decoding stage, which brings improvements in areas such as halluciantions.

## 3 PROPOSED METHOD

### 3.1 OVERVIEW

As illustrated in Fig.2, our method proposes an end-to-end pipeline to mitigate hallucination. Given an input image, we use a VLM to produce an initial caption, which may include hallucinated objects

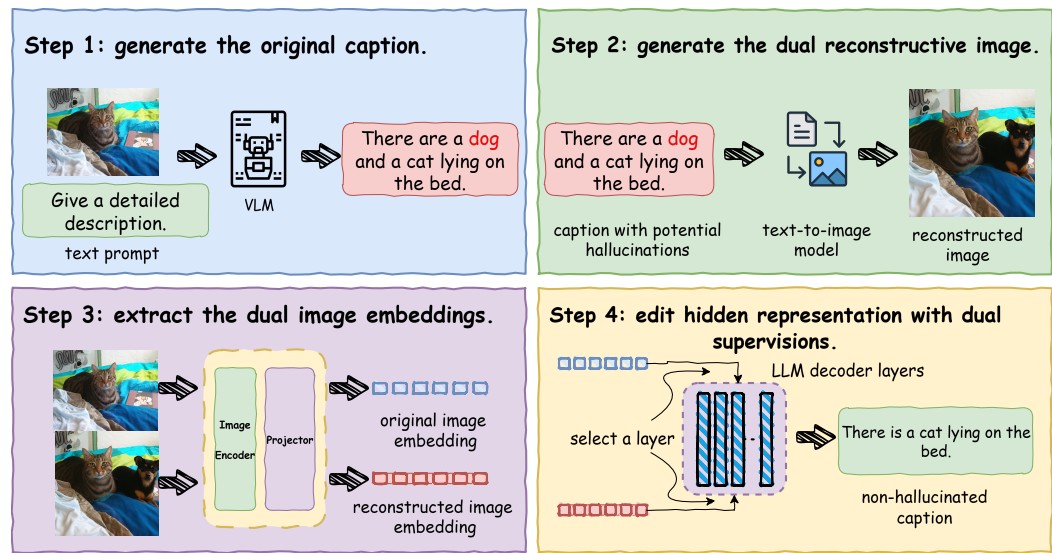

Figure 2: Overview of the proposed method. Given an image and a query, the VLM generates a caption that may contain hallucinations. The caption is then fed into a T2I model to reconstruct an auxiliary image, which amplifies potential hallucinations. Next, both the original and reconstructed images are encoded into embeddings that serve as dual anchors. By injecting these embeddings into the decoder layers to edit hidden representations during inference, the model produces captions that are faithful to visual content without sacrificing informativeness.

or relations. To expose the potential hallucinations in captions, we synthesize a reconstructed image based on the caption using a text-to-image (T2I) model. This reconstruction can naturally amplify and externalize hallucinated content into the visual space. As a result, hallucinations that were originally implicit and difficult to detect in textual semantic space become perceptible once projected into images. Both the original image and the reconstructed image are fed through the image encoder and projection head to obtain the embeddings, denoted as $f(\boldsymbol{I})$ and $f(\boldsymbol{I}')$, respectively. Here, $f(\boldsymbol{I})$ acts as a clean semantic anchor, guiding the representation toward faithful visual semantics, while $f(\boldsymbol{I}')$ explicitly captures the hallucination direction amplified through reconstruction. By simultaneously pulling the image token embeddings toward $f(\boldsymbol{I})$ and pushing them away from $f(\boldsymbol{I}')$, our method establishes an adversarial correction mechanism that requires no hand-crafted metrics or external supervision. Therefore, this design transforms hallucination suppression into a fully self-supervised process, enabling end-to-end correction without human intervention.

## 3.2 HOW TO EXPOSE HALLUCINATIONS

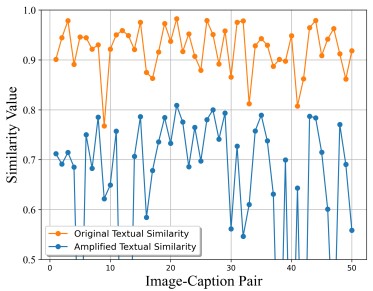

Figure 3: Illustration of the effectiveness of the hallucination amplification mechanism.

Hallucinations in VLMs are intrinsically difficult to detect because they are linguistically well-formed and often indistinguishable from faithful descriptions at the text level. The discrepancy lies not in language plausibility but in the misalignment with visual evidence, which the model itself is typically insensitive to. To address this, we introduce a hallucination exposure mechanism that leverages generative reconstruction to convert implicit inconsistencies into explicit and observable signals. Specifically, given an input image and its generated caption, we use a text-to-image (T2I) model, e.g., FLUX.1-dev (Labs et al., 2025; Labs, 2024), to reconstruct an image from the caption. This reconstructed image exaggerates and materializes the semantics, including hallucinated objects or relations, thereby amplifying otherwise hidden inconsistencies and providing usable supervisory signals for subsequent correction.

To further validate this intuition, Fig.3 illustrates the effectiveness of our hallucination amplification mechanism. For consistency in semantic representation, we compute similarity purely in the textual space. Concretely, let the original image and its caption be denoted as $I$ and $\tau$. By injecting hallucinated information into $\tau$, we obtain a hallucinated caption $\tau'$, which is then fed into a T2I model to reconstruct an image $I'$. Instead of directly comparing the embeddings of $I$ and $I'$, we leverage LLaVA to caption both images, producing $t$ and $t'$, and compute the similarity between them. Thus, we compare the differences between $sim(\tau, \tau')$ and $sim(t, t')$. Notably, even though both the similarity is measured between captions in the textual space, we observe a significant drop once hallucinations are introduced and amplified via reconstruction. This demonstrates that our amplification mechanism transforms subtle, implicit hallucinations into detectable semantic deviations. More importantly, the proposed cross-modal amplification mechanism is entirely training-free and can be applied in a post-hoc manner. It reveals a broader spectrum of hallucination types, including relational and logical inconsistencies, that remain subtle in pure text space, providing an efficient and generalizable pathway.

### 3.3 Hallucination Mitigation via Latent Editing

**Dual Supervision Construction.** We leverage cross-modality reconstruction to amplify potential hallucinations, turning subtle semantic noise in captions into perceptible signals in the visual space. Since our ultimate goal is to mitigate hallucinations in caption generation, it is more effective to operate on the image-space latent representations. We use the original input image $I$ as a semantic anchor, which encodes the clean and faithful semantics. Its embedding, denoted as $f(I)$, obtained through the image encoder and projection head, naturally serves as a ground-truth supervision signal. By pulling the latent representation of the caption closer to this anchor, we ensure that editing does not distort or erase genuine visual information. Meanwhile, the reconstructed image $I'$ is generated based on the obtained caption, which may originally contain hallucinated information. Its embedding $f(I')$ thus provides potentially negative supervision signals, pointing to the direction in the latent space that corresponds to potential hallucinations. By pushing the caption representation away from this embedding, we discourage it from retaining hallucination-related features.

**Latent Representation Editing.** Previous studies (Huh et al., 2024; Jiang et al., 2025; Liu et al., 2025; Stolfo et al., 2025; Jiang et al., 2024) have shown that steering and editing latent representation can guide the generation of LLMs, and have been applied to various tasks such as instruction following and hallucination mitigation, with correction directions typically from handcrafted priors or training objectives. In contrast, our proposed method introduces a training-free and dual-supervision scheme. We combine these two directions and edit the image tokens in the embedding extracted from a selected decoder layer as follows:

$$K'_{h,l} = K_{h,l} + \alpha f(I) - \beta f(I'), \ h \in \mathcal{H}_{img}, \ l \in [1, L], \tag{1}$$

where $K_{h,l}$ stands for the embedding of the $h$-th token at the $l$-th decoder layer, $\mathcal{H}_{img} = \{i_1, i_1 + 1, \ldots i_n\}$ represents the set of positions corresponding to the image-related tokens, $\alpha$ and $\beta$ are scalar coefficients controlling the contributions from the original image $I$ and the dual image $I'$, respectively, and $f(\cdot)$ denotes the combined transformation of image encoder and projector.

Since the potential hallucination signals are derived directly from the model's outputs without requiring external annotation, and the supervision employed for editing is also automatically constructed within the same pipeline, the entire approach constitutes a fully self-supervised, end-to-end knowledge editing strategy that requires no human intervention. Thus, our method can be used as a plug-and-play module in the VLM inference process.

## 4 Experiments

### 4.1 Experimental Setup

**Datasets and Evaluation Metrics.** To comprehensively validate the effectiveness of our method on mitigating hallucination issues on large vision-language models, we utilize the Caption Hallucination Assessment with Image Relevance (CHAIR) (Rohrbach et al., 2019), MLLM Evaluation benchmark (MME) (Fu et al., 2024), and Pooling-based Object Probing Evaluation (POPE) (Li et al., 2023b) benchmarks.

Table 1: Performance of hallucination mitigation on CHAIR across various metrics.

| Methods | CHAIR$_S \downarrow$ | CHAIR$_I \downarrow$ | Average $\downarrow$ | Length$\uparrow$ | Recall $\uparrow$ | $HAR_{@1} \uparrow$ |
|---|---|---|---|---|---|---|
| Baseline | 53.0 | 14.0 | 33.50 | 92.20 | 81.00 | 0.7304 |
| OPERA | **47.8** | 14.6 | 31.20 | 98.65 | 76.80 | 0.7258 |
| ICD | 56.2 | 16.3 | 36.25 | 103.40 | 16.31 | 0.2596 |
| VCD | 48.7 | 14.9 | 31.80 | 100.40 | 77.32 | 0.7247 |
| Ours | **47.8** | **12.7** | **30.25** | 92.44 | 80.10 | **0.7460** |

However, directly using the hallucination rate or recall may not fully assess hallucination mitigation performance. For example, a model can trivially achieve a very low hallucination rate by producing no or overly short captions, but such outputs fail to capture the visual content. To address this, we combine the hallucination rate and the recall into a unified metric and therefore propose the metric $HAR_{@\beta}$ (Hallucination and Recall), which is defined as:

$$HAR_{@\beta} = \frac{(1 + \beta^2 rq)}{\beta^2 q + r}, \ q = 1 - h \tag{2}$$

where $h$ denotes the hallucination rate CHAIR and $r$ denotes the recall ($HAR \in [0, 1]$). The metric is monotonic in both $q$ and $r$, and the parameter $\beta$ controls the trade-off: $\beta > 1$ emphasizes recall, whereas $\beta < 1$ emphasizes reducing hallucinations. In this way, a high $HAR_\beta$ can only be achieved when both the recall and the non-hallucinated rates are high. POPE serves as a complement to CHAIR in object hallucinations on SEEM-annotated datasets, including MSCOCO (Lin et al., 2015), A-OKVQA (Schwenk et al., 2022), and GQA (Hudson & Manning, 2019), thereby broadening the evaluation scope by focusing on the model's performance with context-dependent prompts. Moreover, for hallucination at the attribute level, we adopt the MME benchmark. Similar to POPE, MME tasks are framed as binary Yes-or-No questions, facilitating consistent evaluation. Some results for baseline methods are taken from (Zou et al., 2025).

**Experiment Settings.** We conduct experiments using several of the most representative model architectures. Unless specified, we adopt LLaVA-v1.5-7B (Liu et al., 2023), a widely adopted large vision-language model for image–text understanding and captioning, and FLUX.1-dev, a popular open-source generative model that exemplifies recent advances in text-to-image generation. For fairness, in our comparison, we set all the decoding hyperparameters and temperature the same across different decoding methods. Closely following (Jiang et al., 2025), in our method, we also conduct latent editing in the second layer and uniformly set both $\alpha$ and $\beta$ to 0.1.

## 4.2 HALLUCINATION MITIGATION RESULTS

In this section, we evaluate the model on three benchmarks, i.e., CHAIR, MME, and POPE, to assess its performance in image captioning, object hallucination, and attribute hallucination.

**Results on CHAIR.** As shown in Table 1, our method consistently outperforms other baselines on both CHAIR$_S$ and CHAIR$_I$, demonstrating its superior effectiveness in suppressing hallucinations. Meanwhile, although nearly all methods inevitably reduce recall while suppressing hallucinations, reflecting a trade-off between faithfulness and informativeness, our approach achieves the smallest drop. This demonstrates that our method captures a broad range of ground-truth objects. With the $HAR_{@\beta}$ metric, our method achieves the highest score, highlighting its ability to reduce hallucinations while maintaining coverage. This superior performance originates from our dual-supervision construction, which simultaneously anchors clean semantics from the original image and suppresses hallucination directions from the reconstructed image. In essence, this editing removes only the component along this direction rather than globally suppressing the representation. Thus, our method mitigates hallucinations with little side effect, preserving both informativeness and semantic richness.

**Results on POPE.** While CHAIR primarily evaluates caption hallucinations at the object level, POPE complements it with a polling-based framework. The performance on the POPE benchmark under random, popular, and adversarial settings is presented in Table 2. It can be observed that our method consistently achieves the best performance across all settings. Notably, our method can achieve up to +5.95% accuracy and +6.85% F1 score on average, outperforming other training-free

Table 2: Performance comparisons on POPE across different settings and datasets.

| Dataset | Methods | Random | | Popular | | Adversarial | | Average | |
|---|---|---|---|---|---|---|---|---|---|
| | | Accuracy ↑ | F1-score ↑ | Accuracy ↑ | F1-score ↑ | Accuracy ↑ | F1-score ↑ | Accuracy ↑ | F1-score ↑ |
| MSCOCO | Baseline | 83.49 | 82.28 | 79.98 | 79.34 | 76.03 | 76.26 | 79.83 | 79.29 |
| | ICD | 84.87 | 83.27 | 82.93 | 81.45 | 81.07 | 79.96 | 82.96 | 81.56 |
| | VCD | 86.84 | 86.83 | 82.65 | 83.37 | 77.31 | 79.28 | 82.27 | 83.16 |
| | OPERA | 87.53 | 86.45 | 84.21 | 83.50 | 80.88 | 80.69 | 84.21 | 83.55 |
| | Ours | **89.35** | **89.43** | **86.00** | **86.14** | **81.76** | **82.84** | **85.70** | **86.14** |
| A-OKVQA | Baseline | 83.45 | 82.56 | 79.90 | 79.59 | 74.04 | 75.15 | 79.13 | 79.10 |
| | ICD | 85.57 | 85.06 | 81.93 | 81.95 | 77.43 | 78.99 | 81.64 | 82.00 |
| | VCD | 86.15 | 86.34 | 81.85 | 82.82 | 74.97 | 77.73 | 80.99 | 82.30 |
| | OPERA | 88.27 | 87.54 | 85.17 | 84.74 | **79.37** | 79.97 | 84.27 | 84.08 |
| | Ours | **89.53** | **89.31** | **86.50** | **86.63** | 79.20 | **80.79** | **85.08** | **85.58** |
| GQA | Baseline | 83.73 | 82.95 | 78.17 | 78.37 | 75.08 | 76.06 | 78.99 | 79.13 |
| | ICD | 84.90 | 84.22 | 78.37 | 78.81 | 75.97 | 76.93 | 79.75 | 79.99 |
| | VCD | 86.65 | 86.99 | 80.73 | 82.24 | 76.09 | 78.78 | 81.16 | 82.67 |
| | OPERA | 83.73 | 82.95 | 78.17 | 78.37 | 75.08 | 76.06 | 78.99 | 79.13 |
| | Ours | **87.80** | **87.43** | **82.73** | **83.01** | **79.73** | **80.66** | **83.42** | **83.70** |

Table 3: Performance comparisons on MME.

| Methods | MME-Hall Total ↑ | Object-Level | | Attribute-Level | |
|---|---|---|---|---|---|
| | | Existence ↑ | Count ↑ | Position ↑ | Color ↑ |
| Baseline | 643.3 | 190.0 | 155.0 | 128.3 | 170.0 |
| ICD | 583.3 | 185.0 | 130.0 | 121.7 | 146.7 |
| VCD | 648.3 | 190.0 | 155.0 | 133.3 | **170.0** |
| OPERA | 610.0 | **195.0** | 128.3 | 121.7 | 165.0 |
| Ours | **667.3** | 193.0 | **156.7** | **150.0** | 167.7 |

Table 4: Robustness to hallucination-free captions across different VLMs.

| | Δ of CHAIR | Δ of Recall |
|---|---|---|
| LLaVA-v1.5-7B | 0 | -2 |
| LLaVA-NEXT-7B | 0 | 0 |
| Cambrian-8B | 0 | -2 |
| InstructBLIP-7B | 0 | +1 |

approaches by a large margin. Therefore, these results demonstrate that our method provides a reliable and generalizable solution across different levels of difficulty.

**Results on MME.** In addition to object-level hallucinations, we further evaluate our method on the MME benchmark, which targets attribute-level hallucinations, which is a finer-grained and considerably more challenging setting. It can be observed from Table 3 that our method achieves strong performance on the MME benchmark. In particular, it achieves notable gains on position- and count-related tasks, while maintaining competitive results on object-level existence and color recognition. Moreover, we observe a slight drop in color-related tasks, which can be attributed to the inherent limitations of T2I models in handling fine-grained color information. This phenomenon suggests that the reconstructed images, while effective in amplifying and exposing hallucination directions, may introduce uncertainty in some more fine-grained perception, thereby influencing the internal knowledge of the VLMs. We further discuss this point in Appendix A.

**Cross-Architecture Generalization.** We further employ our method to other widely-used VLM, including LLaVA-NEXT-7B (Liu et al., 2024b), Cambrian-8B (Tong et al., 2024), and InstructBLIP-7B (Dai et al., 2023). As shown in Table 5, our approach demonstrates consistent improvements in hallucination mitigation across different architectures. Notably, on LLaVA-NEXT-7B, our method substantially reduces hallucinations, with $CHAIR_S$ decreasing from 22.4 to 6.6 and $CHAIR_I$ from 6.0 to 2.9. On InstructBLIP-7B, our method not only reduces hallucinations but also improves recall compared to the baseline. Importantly, these gains are achieved without sacrificing caption length or informativeness, striking a favorable balance between faithfulness and coverage. Overall, the results highlight that our proposed dual-supervision editing mechanism can serve as a plug-and-play module across diverse VLMs.

**Robustness Under Non-Hallucinated Conditions** An important property of hallucination mitigation is its robustness: while reducing hallucinations is desirable, the method should not degrade captions that contain no hallucinations. While most existing methods do not evaluate robustness on hallucination-free cases, we explicitly test this by applying our editing mechanism to captions without hallucinations and comparing the outputs before and after latent editing. As shown in Table 4, we report the average change of CHAIR and recall per caption across different models. The results show that the CHAIR metric remains unchanged while recall presents a slight fluctuation ($\pm2$ on average). The results demonstrate that our editing is robust: it suppresses hallucinations when they exist, yet does not introduce side effects when hallucinations are absent. As a result, our method can be seamlessly plugged into different VLMs without the need to first detect hallucinations, highlighting the practical significance.

Table 5: Cross-architecture generalization of our method on LLaVA-NEXT-7B, Cambrian-8B, and InstructBLIP-7B.

| | | CHAIR | | | | | |
|---|---|---|---|---|---|---|---|
| | Method | $CHAIR_S$ | $CHAIR_I$ | Average | Length | Recall | $HAR_{@1} \uparrow$ |
| LLaVA-NEXT-7B | Baseline | 22.4 | 6.0 | 14.20 | 167.78 | 64.40 | 0.7358 |
| | Ours | **6.6** | **2.9** | **4.75** | 119.51 | 60.21 | **0.7377** |
| Cambrian-8B | Baseline | 9.6 | 3.8 | 6.70 | 65.45 | 53.42 | 0.6792 |
| | Ours | **8.0** | **2.9** | **5.45** | 67.43 | 53.29 | **0.6809** |
| InstructBLIP-7B | Baseline | 57.4 | **15.5** | 36.45 | 98.08 | 74.70 | 0.6868 |
| | Ours | **56.4** | 15.8 | **36.10** | 98.61 | 75.17 | **0.6905** |

Table 6: Analytical results on weight factor $\alpha$ and $\beta$ across different settings.

| | $\alpha$ | $\beta$ | $CHAIR_S$ | $CHAIR_I$ | Average | Length | Recall | $HAR_{@1} \uparrow$ |
|---|---|---|---|---|---|---|---|---|
| | 0.09 | 0.09 | 51.8 | 13.9 | 32.85 | 92.33 | 80.18 | 0.7310 |
| | 0.10 | 0.10 | 47.8 | 12.7 | 30.25 | 92.44 | 80.05 | 0.7456 |
| $\alpha = \beta$ | 0.11 | 0.11 | 42.4 | 11.8 | 27.10 | 97.66 | 77.49 | 0.7511 |
| | 0.12 | 0.12 | 40.2 | 12.7 | 26.45 | 97.38 | 75.98 | 0.7460 |
| | 0.08 | 0.12 | 38.6 | 12.4 | 25.50 | 101.80 | 74.48 | 0.7448 |
| | 0.09 | 0.11 | 41.0 | 11.9 | 26.45 | 100.02 | 76.90 | 0.7519 |
| $\alpha \neq \beta$ | 0.11 | 0.09 | 51.2 | 13.5 | 32.35 | 92.32 | 80.91 | 0.7368 |
| | 0.12 | 0.08 | 52.0 | 14.0 | 33.00 | 92.33 | 81.69 | 0.7362 |

## 4.3 ANALYTICAL RESULTS

**Effect of Weight Factor $\alpha$ and $\beta$.** We directly set the weight factors in Eq.1 following (Jiang et al., 2025). In this section, to investigate the influence of weight factors $\alpha$ and $\beta$ in our dual-supervision editing, we conduct experiments under both symmetric ($\alpha = \beta$) and asymmetric ($\alpha \neq \beta$) settings. As shown in Table 6, under the symmetric setting, increasing $\alpha$ and $\beta$ progressively suppresses hallucinations. However, recall simultaneously drops from 80.18 to 75.98 while the response length increases. This shows that larger weights enforce stronger hallucination suppression, but at the cost of reduced content coverage and more verbose outputs. Moreover, the asymmetric setting reveals that encouraging negative supervision, i.e., $\beta > \alpha$, achieves a greater hallucination mitigation while the model is also overly conservative and tends to produce repetitive responses. In contrast, when $\beta$ is smaller than $\alpha$, recall increases, but this comes at the expense of higher hallucination rates. Therefore, the symmetric setting strikes a favorable trade-off, delivering effective hallucination suppression while preserving recall and avoiding overly conservative behavior.

**Effect of Dual Supervision Signals.** To better understand the effect of each supervision signal in Eq.1, we conduct ablation studies. As shown in Table 7, removing both signals represents the baseline performance, which results in the highest hallucination rates. Introducing only the negative supervision $-\beta f(I')$ presents very strong effectiveness in hallucination mitigation and increases the response length. However, we find that in this case, the model behaves overly conservatively: a substantial portion of the responses are repetitive or focus solely on captioning a single object. Thus, the recall is the lowest among variants. While the response length increases, the information richness does not increase, indicating degraded captioning capability. In contrast, introducing only the positive supervision $+\alpha f(I)$ emphasizes the visual tokens of the original image. Since this variant does not introduce any component for hallucination mitigation, it brings limited influence on hallucination performance. When combining both components, our method achieves the best trade-off overall, e.g., achieving the highest $HAR_{@1}$ score. Specifically, since the dual supervision signals influence model performance in different directions, the results deteriorate when either one is absent. For instance, the presence of $f(I')$ tends to make the model overly conservative: while it suppresses fabricated content, it also prevents the model from recognizing objects that actually exist. In contrast, the presence of $f(I)$ serves as a compensation for this drawback. Therefore, integrating $+f(I)$ with $-f(I')$ allows the model to preserve information richness and coverage,

Table 7: Ablation study of the two supervision components in our algorithm using LLaVA-1.5.

| $+\alpha f(I)$ | $-\beta f(I')$ | $CHAIR_S$ | $CHAIR_I$ | Average | Length | Recall | $HAR_{@1}\uparrow$ |
|---|---|---|---|---|---|---|---|
| ✗ | ✗ | 53.0 | 14.0 | 33.50 | 92.20 | 81.00 | 0.7300 |
| ✓ | ✗ | 51.4 | 14.2 | 32.80 | 91.46 | 80.61 | 0.7330 |
| ✗ | ✓ | 46.4 | 12.0 | 29.20 | 101.42 | 76.62 | 0.7360 |
| ✓ | ✓ | 47.8 | 12.7 | 30.25 | 92.41 | 80.05 | 0.7456 |

Table 8: Ablation study of the decoder layers used for latent editing used in our method.

| $l$ | $CHAIR_S$ | $CHAIR_I$ | Average | Length | Recall | $HAR_{@1}\uparrow$ |
|---|---|---|---|---|---|---|
| - | 53.0 | 14.0 | 33.50 | 92.20 | 81.00 | 0.7300 |
| 3 | 47.8 | 12.7 | 30.25 | 92.44 | 80.05 | 0.7456 |
| 6 | 51.6 | 14.3 | 32.95 | 92.16 | 81.50 | 0.7358 |
| 9 | 52.4 | 14.4 | 33.40 | 92.99 | 81.82 | 0.7342 |
| 12 | 50.8 | 13.8 | 32.30 | 91.78 | 80.77 | 0.7365 |
| 15 | 51.0 | 14.1 | 32.55 | 92.38 | 80.45 | 0.7338 |
| 18 | 51.6 | 14.0 | 32.80 | 91.47 | 80.51 | 0.7326 |
| 21 | 51.8 | 13.7 | 32.75 | 91.78 | 81.36 | 0.7364 |
| 24 | 51.0 | 13.4 | 32.20 | 91.72 | 80.45 | 0.7360 |

Table 9: Effect of our proposed hallucination amplification mechanism. We generate the reconstructed image using GPT-image (OpenAI).

| Metric | Baseline | FLUX.1-dev | GPT-image |
|---|---|---|---|
| $CHAIR_S$ | 53.0 | 47.8 | 48.5 |
| $CHAIR_I$ | 14.0 | 12.7 | 13.3 |
| Average | 33.50 | 30.25 | 30.90 |
| Length | 92.20 | 92.44 | 94.01 |
| Recall | 81.00 | 80.05 | 78.70 |
| $HAR_{@1}\uparrow$ | 0.7300 | 0.7456 | 0.7359 |

while still ensuring effective suppression of hallucinations. As a result, removing any component from our method inevitably leads to suboptimal performance.

**Effect of Layer $l$.** Table 8 shows the results of applying our method at different decoder layers. It can be seen that our method consistently proves effective across most layers. Consistent with the results in (Jiang et al., 2025), we observe that intervening at relatively shallow layers yields stronger hallucination suppression, as the injected supervision can influence downstream representations more directly. However, this often comes with a slight drop in recall, suggesting that shallow-layer edits may over-regularize the model's perception. In contrast, applying edits at middle or deeper layers better preserves recall and caption richness while still reducing hallucinations to a certain extent. We will further discuss the layer selection in Appendix A.

**Effect of the Generative Model.** To assess the effect of the hallucination amplification mechanism and the generative models, we evaluate the effectiveness of our method using a different generative model, GPT-image (OpenAI). As shown in Table 9, both variants substantially reduce hallucinations compared to the baseline, achieving lower $CHAIR_S$ and $CHAIR_I$ scores. Importantly, the overall improvements remain consistent across generative models, with only minor variations in recall and response length. This demonstrates that the effectiveness of our approach stems from the amplification mechanism itself rather than any particular generative architecture, highlighting its robustness and generalizability.

## 5 CONCLUSION

This paper proposes a training-free and self-supervised hallucination mitigation method for multimodal large language models, which leverages dual visual anchors to edit the hidden state in an end-to-end manner. Our method can reduce hallucination at the object, attribute, and relation levels without sacrificing informativeness, while showing strong cross-architecture generalization. More importantly, our method achieves superior robustness on hallucination-free data and can serve as a plug-and-play module. We hope that our work inspires further exploration of training-free visual grounding techniques and serves as a practical baseline for building more faithful and reliable VLMs.

## 6 ETHIC AND REPRODUCIBILITY STATEMENT

We introduce a novel method to mitigate hallucination problems in VLMs, improving their safety and reliability for the community. The datasets (benchmarks) used for the evaluation and comparison of our method and baselines are publicly accessible, ensuring the transparency and reproducibility of our work. We will release our work to the community as soon as it is accepted, ensuring that our work is reproduced and grounded for other researchers and practitioners.

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

APPENDIX

## A  DISCUSSION AND FUTURE WORK

We discuss some potential limitations and future work for our method.  1).  Following prior works (Jiang et al., 2025), we conduct latent editing in shallow layers and consistently use the third layer for simplicity. Future work may explore more adaptive strategies for layer selection, such as dynamically identifying the most effective intervention points based on input characteristics or model states. 2). Our method relies on T2I generation to amplify hallucinations. While effective, the performance may be influenced by the fidelity of the T2I model, especially for fine-grained attributes such as color. Future research could integrate stronger generative backbones or develop selective filtering strategies to reduce noise introduced by imperfect reconstructions.  3).  Our primary goal focuses on hallucination mitigation without hurting informativeness, yet this inevitably involves a trade-off.  Future work could explore more principled ways to balance this trade-off, for instance, through adaptive weighting schemes or context-aware editing strategies that tailor the strength of correction to the severity of hallucinations.

Incorporating a T2I model introduces additional latency during inference.  However, it is crucial to contextualize this overhead.  The T2I model is utilized only once to derive the steering vector, and this process does not occur during the per-token decoding phase. The steering operation itself is lightweight at inference.  Therefore, the T2I invocation represents a one-time setup cost rather than a recurring expense, making the overall overhead practical for real-world applications. While the current approach is feasible, we are also exploring methods to further mitigate this initial cost. A promising future direction involves using the distance between the steering vector and the embeddings of a small set of T2I-generated images as a regularization term. This could allow for the optimization of an effective steering vector with substantially reduced computational requirements, further enhancing the efficiency of our method.

While our method achieves strong hallucination mitigation across standard visual scenarios, we acknowledge that scenes containing 30+ objects remain extremely challenging. This limitation largely reflects the current capacity bounds of modern VLMs themselves, which often struggle to provide reliable grounding in such ultra-dense settings. Our approach operates on top of the model's existing capabilities and therefore inherits these fundamental constraints. Extending hallucination mitigation methods to handle highly complex, multi-object scenes is an important and promising future direction.

## B  AI ASSISTANT USAGE STATEMENT

During the preparation of this paper, we made moderate use of Large Language Models (LLMs) for text polishing and for assistance in non-core coding tasks.  However, no LLMs were used in developing the ideas and determining the structure and content of this paper.

## C  DETAILED EXPERIMENTAL SETTINGS

### C.1  POPE

We use the official benchmark from (Li et al., 2023b) work, in which each of the three settings, random, popular, and adversarial, contains 3000 Question-Answer pairs.  The meanings of these three settings as follows: **random** randomly selects non-existent objects to ask about; **popular** selects objects from the top half most frequently appearing in the entire image dataset but not present in the current querying image; **adversarial** first ranks all objects according to their co-occurrence frequencies with the ground-truth objects, and then selects the top-k frequent ones that do not exit in the image. The query template in the benchmark is "Is there a/an [object] in the image?"

### C.2  CHAIR

To ensure the comparability of results, we use the 500 images sampled from MSCOCO by work (Zou et al., 2025). CHAIR is proposed to evaluate object hallucination in image captioning tasks,

which have two variants: per-instance (CHAIR$_I$) and per-sentence (CHAIR$_s$). These are defined as:

$$\text{CHAIR}_I = \frac{|\{\text{hallucinated objects}\}|}{|\{\text{all objects mentioned}\}|}, \ \text{CHAIR}_S = \frac{|\{\text{sentences with hallucinated objects}\}|}{|\{\text{all sentences}\}|}. \quad (3)$$

As mentioned above, $HAR_\beta$ derived from $F_\beta$ score, $F_\beta$ formulates as follows:

$$F_\beta = (1 + \beta^2) \cdot \frac{\text{Precision} \cdot \text{Recall}}{(\beta^2 \cdot \text{Precision}) + \text{Recall}}$$

$$\text{Precision} = \frac{TP}{TP + FP}, \quad \text{Recall} = \frac{TP}{TP + FN}$$

### C.3 MME

The official benchmark from MME (Fu et al., 2024) assesses model performance across 14 diverse vision-language subtasks covering both perception and cognition. Here we select **Existence**, **Count**, **Position**, and **Color** subtasks, which are most associated with the object and attribute hallucinations.

## D   DETAILED EXPERIMENTAL RESULTS

Here we demonstrate two examples of the performance of our method on LLaVA-v1.5-7B. Figure 4 shows the model's performance in the Yes-or-No question is related to its image captioning. We can also observe this relation in Figure 5 that the reconstructed image also contains a potted tree which does not exist.

original image

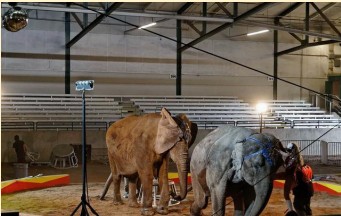

**Prompt:** Is the brown elephant in front of the gray elephant?

**Baseline:** Yes, the brown elephant is in front of the gray elephant.

**Our method:** No, the brown elephant is not in front of the gray elephant. The gray elephant is in front of the brown elephant.

**Original caption:**
The image depicts a group of elephants performing tricks in a circus setting. There are **three** elephants in total, **with one on the left side, another in the middle, and the third on the right side of the scene. The elephants are standing on their hind legs**, showcasing their impressive balance and coordination.
In addition to the elephants, there are **two** people present in the scene. One person can be seen on the left side of the image, while the other person is located on the right side. They are likely the trainers or handlers responsible for guiding the elephants and ensuring their safety during the performance.

reconstructed image

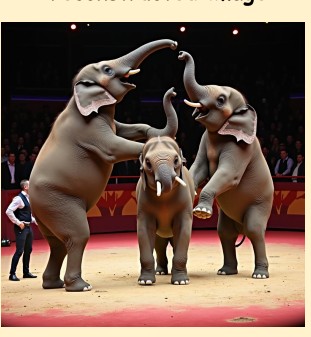

Figure 4: Illustration of comparison between baseline and our method generated by LLaVA-v1.5-7B on MME benchmark.

original image

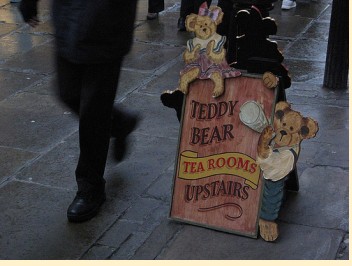

**Prompt:** Is there a tree in the image?

Baseline: Yes, there is a tree.

Our method: No, there is no tree in the image.

**Prompt:** Is there a cardboard in the image?

Baseline: No

Our method: Yes, there is a cardboard sign in the image.

**Original caption:**
The image depicts a **man** walking down a sidewalk in front of a sign that reads \"Teddy Bear Tea Rooms.\" The man is wearing a **black jacket and appears to be the main focus of the scene**.
In addition to the man, there are several other people in the background, some of whom are closer to the foreground, while others are further away. There are also two teddy bears in the scene, one located near the center of the image and the other towards the right side. The presence of these teddy bears adds a playful and whimsical atmosphere to the scene.

reconstructed image

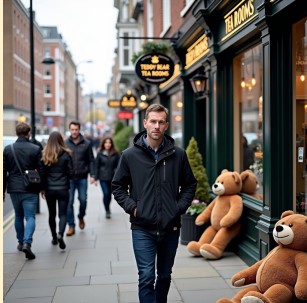

Figure 5: Illustration of comparison between baseline and our method generated by LLaVA-v1.5-7B on POPE benchmark.

Table 10: Effect of the choice of layer in latent editing across different architectures.

|  | $l$ | CHAIR$_S$ | CHAIR$_I$ | Average | Length | Recall | $HAR_{@1}$ ↑ |
|---|---|---|---|---|---|---|---|
| LLaVA-NEXT-7B | baseline | 22.4 | 6.0 | 14.2 | 167.78 | 64.4 | 0.7358 |
|  | 3 | 6.6 | 2.9 | 4.75 | 119.51 | 60.2 | 0.7377 |
|  | 16 | 23.8 | 6.5 | 15.15 | 172.32 | 63.8 | 0.7283 |
|  | 24 | 23.2 | 6.2 | 14.7 | 174.81 | 64.4 | 0.7339 |
| Cambrian-8B | baseline | 9.6 | 3.8 | 6.7 | 65.45 | 53.4 | 0.6792 |
|  | 3 | 8.0 | 2.9 | 5.45 | 67.43 | 52.2 | 0.6726 |
|  | 16 | 11.4 | 4.1 | 7.75 | 65.54 | 53.8 | 0.6796 |
|  | 26 | 11.4 | 4.5 | 7.95 | 66.36 | 52.4 | 0.6678 |
| InstructBLIP-7B | baseline | 57.4 | 15.5 | 36.45 | 98.08 | 74.7 | 0.6868 |
|  | 3 | 56.4 | 15.8 | 36.1 | 98.61 | 74.3 | 0.6871 |
|  | 16 | 54.2 | 15.4 | 34.8 | 97.93 | 75.1 | 0.6980 |
|  | 26 | 58.8 | 16.1 | 37.45 | 98.15 | 75.1 | 0.6825 |

# E  MORE RESULTS ON OTHER BENCHMARKS

Here we frist conduct a comparative experiment on POPE and MME benchmarks with ConVis Park et al. (2024), which is a trainig-free method leveraging T2I models as well (refer to Table 11 12). From the results, we observe that our method outperforms others in both accuracy and time efficiency. **Although both approaches leverage T2I models, our method requires reconstructing only a single image, whereas ConVis must repeatedly reconstruct multiple images, which severely limits the practicality and feasibility of ConVis in real-world deployment.** Moreover, We also evalute our method on more comprehensive benchmarks, i.e., GAVIE Liu et al. (2024a), FaithScoreJing et al. (2024), MMBench Liu et al. (2024d), to assess the perfomance on relation hallucination and more general tasks (refer to Table13 and Table14).

To further demonstrate the stability of our method when using different T2I models, we additionally present the results of our approach on DALL-E3 OpenAI (2023a) and Hyper-SD Ren et al. (2024) (refer to Table 16).

We also compared our method on CHAIR with other models (PerturboLLaVA Chen et al. (2025) and RLAIF-V Yu et al. (2025)) trained using SFT or RL, as well as with stronger training-free methodsLiu et al. (2024c). Considering both hallucination reduction and recall on CHAIR, it can be seen that our method achieves quite a satisfactory performance(refer to Table 15).

Table 11: Comparison between ConVis and our method on POPE and MME benchmarks under the default settings from ConVis.

| | POPE | | MME | | | | |
|---|---|---|---|---|---|---|---|
| | Accuracy | F1-score | Existence | Count | Position | Color | Total |
| baseline | 79.83 | 79.29 | 190 | 153.3 | 133.3 | 155.0 | 643.3 |
| ConVis | - | 83.00 | **195** | **158.3** | 133.3 | 155.0 | 653.6 |
| Ours | **85.70** | **86.14** | 193 | 156.7 | **150.0** | **167.7** | **667.3** |

Table 12: Time overhead comparison between ConVis and our method. The numbers in the table represent the average time the model takes to infer a single instance.(second per instance)

| | POPE | MME |
|---|---|---|
| ConVis | 9.792s | 9.951s |
| Ours | **8.073s** | **8.115s** |

Table 13: Performance of our method on GAVIE and FairhScore benchmarks.

| | GAVIE | | FaithScore | |
|---|---|---|---|---|
| | Accuracy | Relevancy | FaithScore | FaithScore in Sentence |
| baseline | 4.360 | 6.110 | 0.88 | 0.62 |
| Ours | **5.635** | **6.215** | **0.89** | **0.70** |

Table 14: Performance of our method on MMBench.

| | Overall | CP | FP-S | FP-C | AR | LR | RR |
|---|---|---|---|---|---|---|---|
| baseline | 50.95 | 53.32 | 56.28 | 47.37 | 60.76 | 21.39 | **50.71** |
| Ours | **58.97** | **69.59** | **60.05** | **51.42** | **74.31** | **31.21** | 44.08 |

Table 15: Comparison between our method and a range of baselines, including SFT- or RL-trained models as well as recent strong training-free approaches, on the CHAIR benchmark.

| | $CHAIR_S$ | $CHAIR_I$ | Recall | HAR |
|---|---|---|---|---|
| baseline | 53.0 | 14.0 | 81.0 | 0.7304 |
| PertuboLLaVA | 36.1 | 10.4 | - | - |
| VTI | 35.8 | 11.1 | 76.8 | 0.7667 |
| RLAIF-V | 18.1 | 4.7 | 59.2 | 0.710 |
| Ours | 40.8 | 7.5 | **82.0** | **0.7881** |

Table 16: Performance of our method using different T2I models on CHAIR.

|           | $CHAIR_S$ | $CHAIR_I$ | Recall |
|-----------|-----------|-----------|--------|
| baseline  | 53.0      | 14.0      | 81.0   |
| Hyper-SD  | 46.2      | 11.9      | 79.7   |
| DALLE-3   | 49.9      | 12.8      | 78.8   |

# F   MORE RESULTS ON OTHER VLMS ACROSS MORE LAYERS

Table 10 shows the performance of different models at various layers on CHAIR. We can observe that hallucinations are reduced in the shallow layers (3rd layer), while recall is well preserved. Table 18 demonstrates the CHAIR metric of our method applied to different layers of LLaVA-v1.5-7B. We can see that our method is effective when applied to most layers, and its best performance appears at the 3rd layer, possibly due to certain particularities, as other models also show strong hallucination mitigation at this layer.

# G   ROBUSTNESS

An intuitive idea is that the choice of $\alpha$ and $\beta$ should ideally depend on the input image and prompt, since the model exhibits different sensitivity to different inputs. At present, it is infeasible to determine the optimal parameter combination for a specific input. However, we can adopt parameter sampling strategies to explore better-performing values. Specifically, we sample from both uniform and Gaussian distributions(the average of $n$ samples), and evaluate the results under the conditions of $\alpha = \beta$ and $\alpha \neq \beta$, as shown in Table 17. Note that a star indicates results obtained by repeating sampling five times and picking the best. Figure 6 shows the distributions of $\alpha$ and $\beta$ corresponding to the starred results. We can observe that the parameter distributions are relatively uniform, which further demonstrates that our method does not require specific parameter designs. In fact, parameters randomly chosen within an appropriate range can effectively achieve hallucination mitigation.

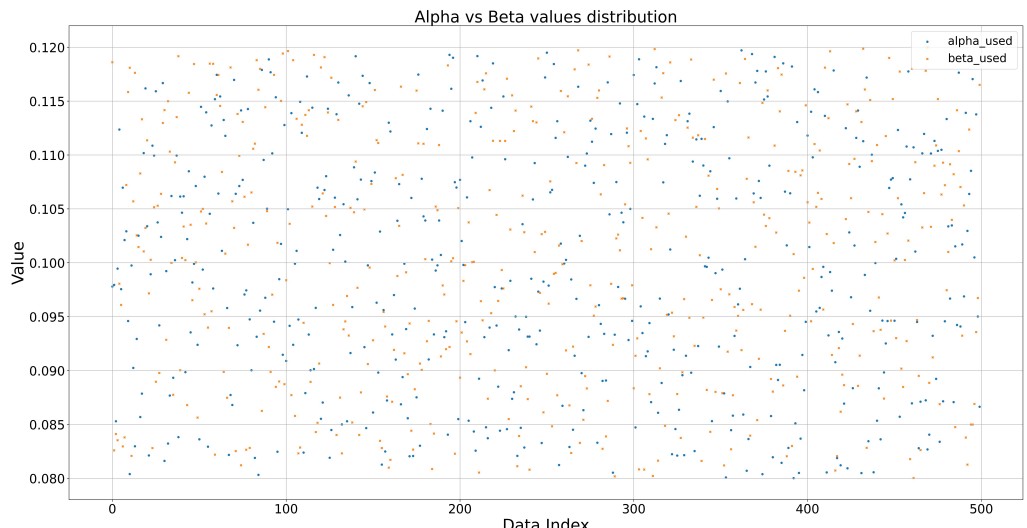

Figure 6: scatter plot of the $\alpha$ and $\beta$ obtained by randomly sampling five times from a uniform distribution and picking the best result, which shows that they are overall uniformly distributed within the range of 0.08 to 0.12.

Table 17: More parameter selection approaches on $\alpha$ and $\beta$. Here, $\mathcal{U}$ denotes sampling from a uniform distribution, $\mathcal{N}$ denotes sampling from a Gaussian distribution, and entries with a star (*) indicate sampling five times and reporting the best result.

| | $\alpha$ | $\beta$ | CHAIR | | | | | |
| --- | --- | --- | --- | --- | --- | --- | --- | --- |
| | | | $CHAIR_S$ | $CHAIR_I$ | Average | Length | Recall | $HAR_{@1}\uparrow$ |
| | 0 | 0 | 53.0 | 14.0 | 33.5 | 92.2 | 81.0 | 0.7304 |
| | 0.1 | 0.1 | 47.8 | 12.65 | 30.23 | 92.4 | 80.05 | 0.7456 |
| $\alpha=\beta$ | $\mathcal{U}(0.08,0.12)$ | $\mathcal{U}(0.08,0.12)$ | 47.0 | 12.4 | 29.7 | 101.8 | 79.1 | 0.7444 |
| | $\mathcal{N}(0.08,0.12)$ | $\mathcal{N}(0.08,0.12)$ | 47.4 | 12.2 | 29.8 | 100.9 | 79.9 | 0.7474 |
| | $\mathcal{U}(0.08,0.12)^\star$ | $\mathcal{U}(0.08,0.12)^\star$ | 40.8 | 7.5 | 24.15 | 109.81 | 82.0 | 0.7881 |
| | $\mathcal{U}(0.08,0.12)$ | $\mathcal{U}(0.08,0.12)$ | 48.6 | 14.4 | 31.5 | 90.7 | 78.2 | 0.7303 |
| $\alpha\neq\beta$ | $\mathcal{N}(0.08,0.12)$ | $\mathcal{N}(0.08,0.12)$ | 45.4 | 12.8 | 29.1 | 94.8 | 78.5 | 0.7451 |
| | $\mathcal{N}(0.08,0.12)^\star$ | $\mathcal{N}(0.08,0.12)^\star$ | 40.6 | 7.1 | 23.85 | 114.0 | 80.5 | 0.7826 |

Table 18: Ablation results on the layers to edit

| $l$ | LLaVA-1.5 | | | | | |
|---|---|---|---|---|---|---|
| | CHAIR$_S$ | CHAIR$_I$ | Average | Length | Recall | $HAR_{@1}\uparrow$ |
| 1 | 53.0 | 14.81 | 33.91 | 91.97 | 79.46 | 0.7216 |
| 2 | 52.0 | 13.93 | 32.97 | 92.81 | 80.45 | 0.7313 |
| 3 | 47.8 | 12.65 | 30.23 | 92.44 | 80.05 | 0.7456 |
| 4 | 53.6 | 14.50 | 34.05 | 91.12 | 81.82 | 0.7303 |
| 5 | 52.0 | 14.29 | 33.15 | 92.54 | 81.89 | 0.7361 |
| 6 | 51.6 | 14.25 | 32.93 | 92.16 | 81.50 | 0.7358 |
| 7 | 50.8 | 14.24 | 32.52 | 90.85 | 80.77 | 0.7171 |
| 8 | 57.2 | 15.52 | 36.36 | 94.90 | 81.23 | 0.7137 |
| 9 | 52.4 | 14.43 | 33.42 | 92.99 | 81.82 | 0.7342 |
| 10 | 52.4 | 14.75 | 33.58 | 92.53 | 81.10 | 0.7303 |
| 11 | 51.2 | 14.27 | 32.74 | 91.77 | 81.17 | 0.7356 |
| 12 | 50.8 | 13.84 | 32.32 | 91.78 | 80.77 | 0.7365 |
| 13 | 51.0 | 14.03 | 32.52 | 93.73 | 81.56 | 0.7385 |
| 14 | 53.0 | 14.30 | 33.65 | 93.33 | 80.38 | 0.7269 |
| 15 | 51.0 | 14.10 | 32.55 | 92.38 | 80.45 | 0.7338 |
| 16 | 50.4 | 13.84 | 32.12 | 91.61 | 81.10 | 0.7390 |
| 17 | 51.2 | 13.77 | 32.49 | 92.01 | 80.45 | 0.7341 |
| 18 | 51.6 | 13.99 | 32.80 | 91.47 | 80.51 | 0.7326 |
| 19 | 51.2 | 14.01 | 32.61 | 91.37 | 80.45 | 0.7334 |
| 20 | 50.4 | 13.92 | 32.16 | 91.62 | 80.97 | 0.7383 |
| 21 | 51.8 | 13.69 | 32.75 | 91.78 | 81.36 | 0.7364 |
| 22 | 51.8 | 13.57 | 32.69 | 91.72 | 80.91 | 0.7349 |
| 23 | 51.4 | 13.77 | 32.59 | 92.05 | 80.64 | 0.7343 |
| 24 | 51.0 | 13.36 | 32.18 | 91.72 | 80.45 | 0.7360 |
| 25 | 52.2 | 14.11 | 33.16 | 91.49 | 81.43 | 0.7342 |
| 26 | 50.8 | 13.67 | 32.24 | 90.99 | 80.64 | 0.7364 |
| 27 | 53.6 | 14.15 | 33.88 | 92.13 | 81.23 | 0.7290 |
| 28 | 52.6 | 14.17 | 33.39 | 92.21 | 81.17 | 0.7317 |
| 29 | 52.8 | 14.14 | 33.47 | 92.22 | 81.30 | 0.7318 |
| 30 | 53.4 | 14.14 | 33.77 | 91.77 | 80.84 | 0.7281 |
| 31 | 53.2 | 13.85 | 33.53 | 91.84 | 81.30 | 0.7314 |
| 32 | 53.0 | 13.96 | 33.48 | 92.23 | 81.04 | 0.7307 |

