# OpenReview forum: "Exposing Hallucinations To Suppress Them: VLMs Representation Editing With Generative Anchors"
_ICLR.cc/2026/Conference — Submitted to ICLR 2026_

### Official Review · Reviewer_bvix · 2025-10-28

**Soundness:** 1
**Presentation:** 3
**Contribution:** 1
**Rating:** 2
**Confidence:** 5

**Summary:**

This paper introduces a training-free hallucination mitigation approach for MLLMs.
The proposed method works as follows: First, a caption is generated from the MLLM using the original image. Next, the caption and the original image are used to generate a new image. Then, the differences between the original and synthetic image representations are used to guide the QA tasks. This idea is tested on four MLLMs, namely LLaVA 1.5, InstructBLIP, LLaVA-Next, and Cambrian, and evaluated on CHAIR, POPE, and MME.

**Strengths:**

- The proposed idea of generating an image from the MLLM-generated caption and the original image, and then using the differences in their latent visual representations to mitigate hallucination, seems interesting.
- The paper is well-written and easy to follow.
- This paper addresses object hallucination, which remains a critical problem in MLLMs.

**Weaknesses:**

* The proposed method does not appear to be an efficient, scalable, or general solution:

  * Image generation is a computationally expensive process.
  * While it may work for simple images containing few objects (e.g., around 5-10 objects), it is likely to fail in complex scenes with multiple objects (e.g., 30+ objects).

* Given this approach requires two auxiliary tasks—caption generation and image generation—before producing an answer. This additional cost should be justified by a significant performance gain. The authors do not show any specific study to report the computation overhead and performance gain; Nevertheless the reported gain is very minimal, making it a unsuitable choice in real-world.

* The experimental setup is weak, as it is evaluated only on POPE and CHAIR, which are outdated and do not comprehensively assess hallucination in LVLMs. POPE evaluates object existence on just 500 images and does not account for other forms of hallucination, such as those involving object attributes or relations. Similarly, CHAIR evaluates only 500 MSCOCO images with limited ground-truth information. These benchmarks lack diversity and rigor. The authors are encouraged to use more recent and challenging benchmarks such as HallusionBench, AMBER, M-HalDetect, and GAVIE.

* The paper relies on older MLLMs such as InstructBLIP and LLaVA 1.5, showing only marginal improvements in certain settings, which makes the effectiveness of the proposed method less convincing. To better demonstrate the method’s generality, the authors could evaluate it on more recent models such as Qwen-VL 2.5 or 3, InternVL 3, and BLIP-3. Additionally, several results (e.g., on POPE and MME) are missing for newer models like Cambrian and LLaVA-Next, which are already mentioned in the paper.

* The comparison setup is also weak, the proposed method is only compared against a few old decoding techniques such as VCD and OPERA; I recomend authors to also compared against newer training-free methods, as well as other online/offline RL methods.

**Questions:**

Please see the weaknesses above.

---

> ### Author Response · Authors · 2025-11-21
>
> - **Q1: How do the authors justify the efficiency and scalability of their method, given that it relies on computationally expensive image and caption generation steps, and the reported performance gains appear minimal? It is likely to fail in complex scenes with 30+ objects.**
> - **A1:** Thank you for your question. To validate the efficiency and scalability of our method, we compare our method with another related work, ConVis [a], which also leverages T2I models to reduce model hallucinations. As shown in the tables below, our method achieves more substantial performance gains with stronger efficiency compared to [a]. Thus, our method achieves a better balance between the effectiveness and efficiency.\
> We agree that scenes that contain 30+ objects are extremely challenging. However, this limitation does not stem from our method; current VLMs themselves may fail to perform reliably in such ultra-dense scenarios, which exceed the visual capacity. Extending hallucination mitigation to extremely dense, multi-object scenarios is an important and promising future direction, but out of scope here, and we appreciate the reviewer highlighting this point. We have included this in our discussion on page 15 in the revised manuscript.\
> **Table A-1: Comparison with ConVis using POPE and MME with LLaVA-v1.5-7B**
> | | **POPE** | | **MME** | | | | |
> |:---|---:|---:|---:|---:|---:|---:|---:|
> | | Accuracy | F1-score | Existence | Count | Position | Color | Total |
> | baseline | 79.83 | 79.29 | 190 | 153.3 | 133.3 | 155.0 | 643.3 |
> | ConVis | - | 83.00 | **195** | **158.3** | 133.3 | 155.0 | 653.6 |
> | Ours | **85.70** | **86.14** | 193 | 156.7 | **150.0** | **167.7** | **667.3** |
> **Table A-2: Comparison of training cost with ConVis.**
> | | POPE | MME |
> | ConVis | 9.792s | 9.951s |
> | Ours | **8.073s** | **8.115s** |
>
> [a]  Park, Yeji, et al. "Convis: Contrastive decoding with hallucination visualization for mitigating hallucinations in multimodal large language models." Proceedings of the AAAI Conference on Artificial Intelligence. Vol. 39. No. 6. 2025.
>
> - **Q2: The experiment is only conducted on CHAIR and POPE, and POPE evaluates on just 500. CHAIR evaluates only 500 MSCOCO images. Should introduce newer and recent benchmarks.**
> - **A2:**  Thank you for the insightful comments. We would like to clarify that our experimental settings closely follow prior works [a-d]: POPE and CHAIR remain the standardized and most widely used hallucination benchmarks in recent VLM works.
> Meanwhile, POPE in our paper is conducted on three datasets, MSCOCO, AOKVQA, and GQA, containing more than 500 images. This is consistent with the official POPE protocol. For CHAIR, we rigorously follow the standard 500-image split used in existing works **for a fair comparison**.\
> Following your suggestion, we have conducted more evaluations on newer and broader benchmarks, including GAVIE, FaithScore, and MMBench, which span general vision-language tasks, attributive reasoning, and relational consistency. As shown in the table below, our method also achieves superior effectiveness of hallucination mitigation. These results have been included in Appendix E on page 19  in the revised manuscript.\
> **Table A-3 Performance of our method using GAVIE and FaithScore.**
> | | GAVIE | | FaithScore | |
> |:---|:---:|:---:|:---:|:---:|
> | | Accuracy | Relevancy | Faithscore | Faithscore in sentence |
> | Baseline | 4.360 | 6.110 | 0.88 | 0.62 |
> | Ours | **5.635** | **6.215** | **0.89** | **0.70** |
>
> **Table A-4 Performance of our method on MME Benchmark**
> | | MMBench | | | | | | |
> |:---|:---:|:---:|:---:|:---:|:---:|:---:|:---:|
> | | Overall | CP | FP-S | FP-C | AR | LR | RR |
> | Ours | **58.97** | **69.59** | **60.05** | **51.42** | **74.31** | **31.21** | 44.08 |
> | Baseline | 50.95 | 53.32 | 56.28 | 47.37 | 60.76 | 21.39 | **50.71** |
>
> [a] Zou, Xin, et al. "Look twice before you answer: Memory-space visual retracing for hallucination mitigation in multimodal large language models." Proceedings of the International Conference on Machine Learning, Vancouver, Canda. PMLR 267, 2025.\
> [b] Jiang, Nick, et al. "Interpreting and editing vision-language representations to mitigate hallucinations." Proceedings of International Conference on Learning Representations, Sinapore, 2025.\
> [c] Liu, Sheng, et al. "Reducing hallucinations in vision-language models via latent space steering." Proceedings of International Conference on Learning Representations, Sinapore, 2025.\
> [d] Park, Yeji, et al. "Convis: Contrastive decoding with hallucination visualization for mitigating hallucinations in multimodal large language models." Proceedings of the AAAI Conference on Artificial Intelligence. Vol. 39. No. 6. 2025.\

---

> ### Author Response · Authors · 2025-11-21
>
> - **Q3:To better demonstrate the method’s generality, the authors could evaluate it on more recent models such as Qwen-VL 2.5 or 3, InternVL 3, and BLIP-3. Additionally, several results (e.g., on POPE and MME) are missing for newer models like Cambrian and LLaVA-Next, which are already mentioned in the paper.**
> - **A3:** Thanks for the comments. Importantly, our evaluation covers more VLMs than prior methods (e.g., ConVis[d], ProjectAway[b], VTI[c], ICD[a]), providing a broader and more up-to-date comparison. Our main experiments are conducted on LLaVA-v1.5. We additionally report the performance of several other models (InstructBLIP, Cambrian, LLaVA-NEXT) on CHAIR in the paper to provide an initial demonstration of the generality of our method across different models.\
> **ConVis:** MiniGPT-4, mPLUG-Owl2, LLaVA-v1.5\
> **Projectaway:** InstructBLIP, LLaVA-v1.5, LLaVA-NEXT, Cambrian\
> **VTI:** LLaVA-v1.5, InstructBLIP\
> **ICD:** InstructBLIP, MiniGPT-4\
> [a] Wang, Xintong, et al. "Mitigating hallucinations in large vision-language models with instruction contrastive decoding." Findings of the Association for Computational Linguistics ACL 2024\
> [b] Jiang, Nick, et al. "Interpreting and editing vision-language representations to mitigate hallucinations." Proceedings of International Conference on Learning Representations, Sinapore, 2025.\
> [c] Liu, Sheng, et al. "Reducing hallucinations in vision-language models via latent space steering." Proceedings of International Conference on Learning Representations, Sinapore, 2025.\
> [d] Park, Yeji, et al. "Convis: Contrastive decoding with hallucination visualization for mitigating hallucinations in multimodal large language models." Proceedings of the AAAI Conference on Artificial Intelligence. Vol. 39. No. 6. 2025.
> - **Q4: How can the authors strengthen the comparison setup, given that the proposed method is currently evaluated only against a few older decoding techniques like VCD and OPERA, and not against newer training-free methods or other online/offline RL approaches?**
> - **A4:** In fact, this kind of comparison was made intentionally and conventionally. Most related studies likewise conduct experiment comparison mainly against training-free approaches. Although somewhat unfair, we still present the performance of VTI[a] and RLAIF-V[b] on the CHAIR benchmark here.\
> | | CHAIR_s | CHAIR_i | Recall | HAR |
> |:---|:---:|:---:|:---:|:---:|
> | Baseline | 53.0 | 14.0 | 81.0 | 0.7304 |
> | VTI | 35.8 | 11.1 | 76.8 | 0.7667 |
> | RLAIF-V | 18.1 | 4.7 | 59.2 | 0.710 |
> | Ours | 40.8 | 7.5 | **82.0** | **0.7881** |
>
> [a] Liu, Sheng, et al. "Reducing hallucinations in vision-language models via latent space steering." arXiv preprint arXiv:2410.15778 (2024).\
> [b] Yu, Tianyu, et al. "Rlaif-v: Open-source ai feedback leads to super gpt-4v trustworthiness." Proceedings of the Computer Vision and Pattern Recognition Conference. 2025.

---

> ### Author Response · Authors · 2025-11-23
>
> Dear reviewer bvix:
>
> Thanks so much again for the time and effort in our work. According to the comments and concerns, we conduct the corresponding experiments and further discuss the related points. Additionally, according to your suggestions, we have revised the manuscript and added more experiments in Appendix E.
>
> As the discussion period is about to close, may I know if our rebuttal addresses the concerns? If there are further concerns or questions, we are willing to address them. Thanks again for taking the time to review our work and provide insightful comments.

---

> ### Comment · Reviewer_bvix · 2025-11-25
>
> Dear Authors,
>
> Thank you for your response.
>
> Despite the other limitations, I did like the idea of generating an image from the MLLM-generated caption and the original image, and then using the differences in their latent visual representations to mitigate hallucination. However, as `Rev. jLcd` noted, this idea was first introduced in ConVis, which the authors did not cite or include in their earlier version. This further weakens the contribution. The authors should have marked all revisions in blue so the changes are easy to track.
>
> My major concerns remain the weak experimental setup, limited improvements, and lack of novelty. Therefore, I am sticking to my original score and recommend rejection.

---

> ### Author Response · Authors · 2025-11-29
>
> Thanks for your response.
>
> Below I list all the concerns that you raised in your recent comment.
>
> - **Q1: Weak experimental setup (need more benchmarks, CHAIR lacks diversity, POPE only contains 500 images)**
> -  **A1:** First, the POPE benchmark we used consists of  **1,500** images; your claim that POPE contains only **500** images is **factually incorrect**. Importantly, we faithfully followed the experimental setup used in contemporary work. In our initial version, we had already provided diverse experiments demonstrating our method’s effectiveness. As suggested, we further **added three additional benchmarks as well as comparisons with other two baselines, substantially exceeding** the experimental scope of contemporary work. Yet, **without providing a positive acknowledgment**, you still characterized the experiments as “weak.”
>
> - **Q2: Limited improvement.**
> - **A2:** This is **factually incorrect and unresaonable.** On the CHAIR benchmark, our method yields a **27.91%** reduction in hallucination while maintaining the highest recall score. In comparison, all other baselines reduce hallucination by no more than **10%**, demonstrating a **substantial performance gap**. Even when compared against an RL-based method, RLAIF-V, our training-free techniques still achieve the **best overall trade-off between hallucination reduction and recall**. From a comprehensive view, our method is the SOTA over other baselines in this paper, which are all the **highly cited and classic work** contemporarily.
>
> - **Q3: Lack of novelty.**
> - **A3:** This is **facually incorrect** decision **uncritically**. Our method is the **FIRST** work to empower steering techniques with semantic reconstruction. We propose a novel way to create contrastive signals with Generative Anchor during steering, which is unsupervised and powerful. ConVis and our approach belong to **entirely different**  research sub-domains. What we introduce is the **first** complete application and practical implementation of semantic reconstruction for steering, something that **ConVis does not address at all**.

---

### Official Review · Reviewer_xReZ · 2025-10-31

**Soundness:** 3
**Presentation:** 3
**Contribution:** 3
**Rating:** 6
**Confidence:** 3

**Summary:**

This paper proposes a training-free, self-supervised method to mitigate hallucinations in multimodal large language models. The approach first reconstructs an image from a model-generated caption via a text-to-image model, exposing implicit hallucinations as explicit visual artifacts. It then forms dual visual anchors, the original image as a positive semantic reference and the reconstructed one as a negative signal, to steer the decoder’s hidden states through lightweight latent editing. Extensive experiments across CHAIR, POPE, and MME benchmarks demonstrate consistent hallucination reduction with minimal loss of informativeness and strong cross-architecture generalization. Overall, the method provides a straightforward yet effective plug-and-play solution for enhancing factual grounding in large vision-language models.

**Strengths:**

- The idea of projecting captions into the visual domain to amplify hallucinations is conceptually clear, technically lightweight, and eliminates the need for external labels or detectors.
- Pulling representations toward faithful semantics (+f(I)) while pushing away from hallucination directions (−f(I′)) provides a well-motivated contrastive structure that cleanly maps to latent geometry.
- The approach introduces no additional training cost and integrates smoothly into inference pipelines, making it genuinely deployable in real systems.
- The study carefully dissects α/β weights, supervision signals, and layer depth, providing transparent insight into how each factor shapes hallucination suppression versus recall trade-offs.

**Weaknesses:**

- The success of hallucination amplification hinges on the generative quality of the chosen T2I model; biases or artifacts in reconstruction could introduce spurious “negative” directions.
- While intuitively appealing, the paper lacks a formal justification or analysis of why linear editing in hidden space reliably aligns with semantic hallucination dimensions.
- Since the negative anchor is derived from the model’s own (possibly erroneous) caption, the supervision may reinforce existing language-model biases rather than objectively correct them.
- Reported improvements are modest and could fall within evaluation noise; statistical significance or variance analysis is missing.

**Questions:**

- How sensitive is the method to the specific T2I model used—would weaker backbones still expose hallucinations effectively?
- Does applying latent editing at multiple decoder layers cumulatively improve results, or does interference occur between layers?
- Did you conduct any statistical significance tests to verify the robustness of your experimental results?

---

> ### Author Response · Authors · 2025-11-20
>
> - **Q1: Bias or artifacts from T2I model could introduce spurious “negative” directions.**
> - **A1:** Thank you for the insightful question. As shown in Table 9, we conducted an ablation study using GPT-Image and observed that the performance remained consistently strong. To further examine the effect of potential biases or artifacts in T2I models, we additionally evaluated two other T2I models. Their results differ only slightly from one another, indicating that our method is **robust across different T2I sources**. Notably, even **DALLE-3**, an older generation T2I model, still achieves satisfactory performance under our framework.\
> While we acknowledge that T2I models may introduce certain spurious “negative” directions, the empirical evidence suggests that these imperfections do **not** undermine the effectiveness of our approach. On the contrary, despite the current limitations of T2I instruction-following, the method already **significantly mitigates hallucinations in VLMs**, demonstrating strong practical reliability.
> | | Baseline | FLUX.1_dev | GPT-image | Hyper-SDXL | DALL-E3
> |:---|---:|---:|---:|---:|---:|
> | $CHAIR_S$ | 53.0 | 47.8 | 48.6 | 46.2 |49.9|
> | $CHAIR_I$ | 14.0 | 12.6 | 13.2 | 11.9 |12.8|
> | Recall | 81.0 | 80.9 | 78.7 | 79.7 |78.8|
> - **Q2: Lack of rationale of linear editing in hidden space reliably aligns with semantic hallucination dimensions.**
> - **A2:** Thoughtful comments. We would like to clarify that our method builds upon a well-established line of research on steering vectors, rather than assuming linearity without grounding. A substantial body of empirical work has repeatedly demonstrated that high-level semantic factors in LLMs and VLMs exhibit locally linear structure in their hidden representations, enabling linear directions to reliably modulate concepts such as sentiment, style, object attributes, or safety behaviors. Recent studies in VLM likewise show that semantic steering can be effectively implemented through linear operations in the hidden space, and several works have offered hypotheses explaining why such local linearity emerges during training.\
> Our approach follows the same empirical foundation: we identify a direction correlated with hallucination tendencies and apply controlled linear editing along this axis. We have included these in line 243-245 in the revised manuscript.\
> [a] Jiang, Yibo, et al. "On the origins of linear representations in large language models." International Conference on Machine Learning, Vienna, Austria, 2024\
> [b] Jiang, Nick, et al. "Interpreting and editing vision-language representations to mitigate hallucinations." Proceedings of International Conference on Learning Representations, Sinapore, 2025.\
> [c] Liu, Sheng, et al. "Reducing hallucinations in vision-language models via latent space steering." Proceedings of International Conference on Learning Representations, Sinapore, 2025..
> - **Q3: Does using negative anchors derived from the model’s own hallucinated captions risk reinforcing the model’s inherent language biases instead of correcting them?**
> - **A3:** We appreciate the reviewer’s concern. Although the negative anchors are derived from the model’s own hallucinated captions, they do not reinforce the model’s existing biases. Instead, they serve as counter-examples that explicitly push the model away from its hallucination tendencies. Importantly, the supervision is contrastive: the model learns to pull toward the original image semantics while being pushed away from the hallucinated description. This relative signal does not rely on the correctness of the hallucinated caption itself, but only on its difference from the ground-truth visual content.
> In a sentence, we agree the original caption may contain wrong (even not hallucination) information, but it doesn’t matter that we “minus” it which means suppressing it.
> - **Q4/Q5: Reported improvements are modest and the gain may be derived from evaluation noise, statistical significance or variance analysis is missing. Did you conduct any statistical significance tests to verify the robustness of your experimental results?**
> - **A4/A5:**  Thanks for the question. We would like to clarify that the improvements reported in our paper are not marginal and cannot be attributed to evaluation noise. As shown in Table 1 in paper,  our method reduces hallucination by **9.70%** relative to the baseline and we would hit hallucination reduction **27.91%** under our best setting, whereas other concurrent training-free methods (VCD, ICD, OPERA) achieve significantly smaller or even negative changes, with reductions of only 5.07%, -8.21%, and 6.86%, respectively. For the key Recall metric, our method drops only **1.11%**, whereas all other methods suffer drops **>5.19%**, indicating that we effectively suppress hallucinations without harming core model capability. Regarding statistical significance, all results in the paper are already averaged over three independent runs.

---

> ### Author Response · Authors · 2025-11-20
>
> - **Q6: How sensitive is the method to the specific T2I model used — would weaker backbones still expose hallucination effectively?**
> - **A6:** Thank you for the question. As discussed in A1, we evaluated multiple T2I models and observed only **minor performance differences** across them. Notably, even **DALLE-3**, which is relatively old and far weaker than modern T2I models, still produces **satisfactory and stable results** under our framework. This demonstrates that our method does not rely on strong T2I backbones, because even with models like DALL-E 3, whose long-prompt reconstruction ability is not particularly strong, our method still works. As long as the model is capable of exposing the intrinsic hallucination of the VLM, our approach can reduce hallucinations to some extent.
> - **Q7: Does applying latent editing at multiple decoder layers cumulatively improve results, or does interference occur between layers?**
> - **A8:** This is an interesting question! To my knowledge, current steering methods[a][b][c] that perform edits on hidden states operate on a single layer. Therefore, we followed this convention.\
> [a]Jiang, Nick, et al. "Interpreting and editing vision-language representations to mitigate hallucinations." Proceedings of International Conference on Learning Representations, Sinapore, 2025\
> [b]Liu, Sheng, et al. "Reducing hallucinations in vision-language models via latent space steering." Proceedings of International Conference on Learning Representations, Sinapore, 2025.\
> [c]Dunefsky, J., & Cohan, A. (2025). Investigating generalization of one-shot LLM steering vectors. Second Conference on Language Modeling, Montreal, Canada, 2025.

---

> ### Author Response · Authors · 2025-11-23
>
> Dear reviewer xReZ:
>
> Thanks so much again for the time and effort in our work. According to the comments and concerns, we conduct the corresponding experiments and further discuss the related points. Additionally, according to your suggestions, we have revised the manuscript and added more experiments in Appendix E.
>
> As the discussion period is about to close, may I know if our rebuttal addresses the concerns? If there are further concerns or questions, we are willing to address them. Thanks again for taking the time to review our work and provide insightful comments.

---

> > ### Comment · Reviewer_xReZ · 2025-11-27
> >
> > Thank you for the detailed clarification and for revising the manuscript to address the points I raised. I checked the updated sections, and additional explanations have substantially resolved the concerns I previously had.

---

### Official Review · Reviewer_vnXo · 2025-10-31

**Soundness:** 3
**Presentation:** 3
**Contribution:** 3
**Rating:** 6
**Confidence:** 4

**Summary:**

This paper proposes a training-free, self-supervised method to mitigate hallucinations in multimodal large language models. It leverages a hallucination amplification mechanism, using text-to-image projections as negative anchors and real images as positive anchors to correct decoder representations. The approach significantly reduces object-, attribute-, and relation-level hallucinations without finetuning or performance loss. Experiments show strong cross-architecture generalization and minimal side effects on faithful captions.

**Strengths:**

1. The proposed training-free, self-supervised framework effectively mitigates hallucinations in MLLMs without additional finetuning or handcrafted priors, offering an efficient and scalable low-cost solution across object, attribute, and relation levels.
2. The innovative hallucination amplification mechanism with dual visual-text anchors enables fine-grained detection and correction of subtle hallucinations while preserving informativeness and maintaining model reliability.

**Weaknesses:**

1. The paper claims to suppress three types of hallucinations: "object-level, attribute-level, and relation-level". However, experiments are only conducted on CHAIR (object-level), POPE (object-level), and MME (attribute-level) benchmarks, with no dedicated evaluation designed for relation-level hallucinations. The benchmark in [a] should be considered.
2. The paper only compares with a limited number of inference-stage methods, such as OPERA, ICD, and VCD, and does not incorporate mainstream training-stage methods (e.g., hallucination suppression based on RLHF, visual supervision fine-tuning) for comparison. Methods such as PerturboLLaVA [a] and RLAIF-V [b] should be added.
3. The paper fails to clarify the dimensional matching and semantic alignment of the original image embedding (positive anchor) and the reconstructed image embedding. For example, given visual content differences between the original and reconstructed images (e.g., the latter contains hallucinated objects), it remains unaddressed whether their embeddings—generated via the same image encoder and projection head—reside in the same semantic space. If spatial misalignment exists, the validity of the adversarial correction logic ("pulling toward the positive anchor and pushing away from the negative anchor") is also unconfirmed, with no relevant verification or theoretical explanation provided.


[a] Chen, Cong, et al. "PerturboLLaVA: Reducing multimodal hallucinations with perturbative visual training." arXiv preprint arXiv:2503.06486 (2025).
[b] Yu, Tianyu, et al. "Rlaif-v: Open-source ai feedback leads to super gpt-4v trustworthiness." Proceedings of the Computer Vision and Pattern Recognition Conference. 2025.

**Questions:**

See the weaknesses.

---

> ### Author Response · Authors · 2025-11-20
>
> - **Q1: How do the authors justify their claim of suppressing relation-level hallucinations, given that no dedicated evaluation for relation-level hallucinations was conducted, and should the benchmark in “PertorboLLaVA” be included?**
> - **A1:**  Thanks for your comments. To address your comment, we evaluate our method using GAVIE, FaithScore, and MMBench from the suggested PerturboLLaVA. As shown in the tables below, our method achieves superior performance across a broader range of hallucination scenarios.\
> **Table B-1: Performance under GAVIE and FaithScore.**
> | | GAVIE | | FaithScore | |
> |:---|:---:|:---:|:---:|:---:|
> | | Accuracy | Relevancy | Faithscore | Faithscore in sentence |
> | Baseline | 4.360 | 6.110 | 0.88 | 0.62 |
> | Ours | **5.635** | **6.215** | **0.89** | **0.70** |
> **Table B-2:  Performance under MMBench.**
>
> | | Overall | CP | FP-S | FP-C | AR | LR | RR |
> |:---|:---:|:---:|:---:|:---:|:---:|:---:|:---:|
> | Ours | **58.97** | **69.59** | **60.05** | **51.42** | **74.31** | **31.21** | 44.08 |
> | Baseline | 50.95 | 53.32 | 56.28 | 47.37 | 60.76 | 21.39 | **50.71** |
>
> - **Q2: The paper only compares with a limited set of inference-stage methods, without including mainstream training-stage hallucination-suppression approaches such as PerturboLLaVA and RLAIF-V for comparison?**
> - **A2:** Thank you for the question. Our work focuses on **inference-stage hallucination mitigation**, and most concurrent studies likewise compare primarily against **training-free approaches**, as these methods share similar applicability, computational cost, and deployment constraints. In contrast, training-stage hallucination-suppression approaches such as **PerturboLLaVA** and **RLAIF-V** operate under a fundamentally different paradigm. For instance, RLAIF-V requires reinforcement learning with strong computational budgets far beyond our inference-stage setting, making head-to-head comparison **unfair**.\
> Nevertheless, to fully address your comments, we report the available numbers of PerturboLLaVA and RLAIF-V on the CHAIR benchmark. When considering both hallucination reduction and recall, our method still **outperforms** the other approaches in the table. **We have included these results in the appendix E in the revised manuscript.**
> | | $CHAIR_s$ | $CHAIR_I$ | Recall | HAR |
> |:---|:---:|:---:|:---:|:---:|
> | Baseline | 53.0 | 14.0 | 81.0 | 0.7304 |
> | PertuboLLaVA | 36.1 | 10.4 | - | - |
> | RLAIF-V | 18.1 | 4.7 | 59.2 | 0.710 |
> | Ours | 40.8 | 7.5 | **82.0** | **0.7881** |
> - **Q3: How do the authors ensure dimensional and semantic alignment between the embeddings of the original image and the reconstructed image, and what evidence supports the validity of the adversarial correction logic if the reconstructed image may contain hallucinated content?**
> - **A3:** Thank you for your question.
>   - First, regarding the dimension: both the original and the reconstructed image are passed through the model's own image encoder and projector, so the dimension of the embeddings is guaranteed to be the same and is equal to the model's dimension. Regarding semantics: according to the hypothesis in [a], all deep neural networks are learning a certain projection of real concepts in different dimensional spaces. If we consider natural language as the real "concept," then both the VLM's understanding and the T2I model's generation are attempts to reproduce this concept, and this is also the training objective for both. That is why we can seamlessly unite the two using "image caption."
>   - Regarding the issue of spatial misalignment, let me clarify again: the prompt fed into the T2I model corresponding to the image generated by the T2I model is precisely the hallucination produced by the model itself. The T2I model is meant to do its best to exhibit the model's existing misunderstanding so that this error can be well suppressed during the editing process. Admittedly, the T2I model itself may generate additional, unnecessary hallucinations, which depends on the T2I model's instruction-following ability. However, a large amount of solid experimental data shows that the possible negative impact caused by this concern is far less than the gain from correctly eliminating the hallucination while retaining the model's ability to understand the image.
>
> [a] Huh, Minyoung, et al. "The platonic representation hypothesis." International Conference on Machine Learning, Vancouver, Canada, 2025.

---

> ### Author Response · Authors · 2025-11-23
>
> Dear reviewer vnXo:
>
> Thanks so much again for the time and effort in our work. According to the comments and concerns, we conduct the corresponding experiments and further discuss the related points. Additionally, according to your suggestions, we have revised the manuscript and added more experiments in Appendix E.
>
> As the discussion period is about to close, may I know if our rebuttal addresses the concerns? If there are further concerns or questions, we are willing to address them. Thanks again for taking the time to review our work and provide insightful comments.

---

### Official Review · Reviewer_jLcd · 2025-11-01

**Soundness:** 2
**Presentation:** 2
**Contribution:** 1
**Rating:** 2
**Confidence:** 5

**Summary:**

This paper investigates the vulnerability of large vision-language models (LVLMs) to hallucinations triggered by visual perturbations. The authors propose a novel visual adversarial attack framework that subtly modifies images to expose and amplify hallucination behaviors without altering the overall semantics or human perception of the image. Through experiments on several state-of-the-art LVLMs, the study shows that even small, imperceptible visual perturbations can cause severe hallucinations in both captioning and VQA tasks, revealing inconsistencies between visual grounding and textual reasoning. The method provides a systematic way to evaluate the robustness and faithfulness of LVLMs, highlighting the gap between human-aligned perception and model interpretation.

**Strengths:**

* The paper proposes a training-free contrastive decoding method that leverages a text-to-image (T2I) model to mitigate hallucinations in large vision-language models (LVLMs).

* Experimental results demonstrate that the proposed method effectively reduces hallucinations without compromising the general performance of the LVLM, highlighting its practicality as a lightweight, training-free solution.

**Weaknesses:**

* Limited novelty and missing reference. The paper omits a highly relevant prior work, ConVis (Park et al., 2025), which also employs reconstructed images from hallucinated captions to generate negative anchors and performs training-free contrastive decoding. The overlap in both motivation and method design raises concerns about the novelty of the proposed approach. The authors should include ConVis as a key reference and conduct comparative experiments to clearly differentiate their contribution.

* Dependence on the quality of the image generation model. Since the proposed method relies heavily on the T2I model, its performance may vary depending on the model’s alignment and reconstruction quality. If the generated image introduces irrelevant visual cues, the contrastive signal may fail to reduce hallucination effectively. Moreover, the current experiments appear to use a single image generation, which could be influenced by stochastic variation. Running the T2I process multiple times and reporting averaged results would strengthen the reliability of the findings.

* To further validate the effectiveness of the reconstructed image as a negative anchor, it would be beneficial to evaluate the method on additional benchmarks beyond CHAIR and POPE — for example, PhD (Liu et al., 2025) for counter-commonsense reasoning or another benchmark that assesses general performance besides MME.

Minor: The paper inconsistently uses the terms LVLM and MLLM. It would be better to standardize terminology throughout the text.

**References**:
- Park, Yeji, et al. "Convis: Contrastive decoding with hallucination visualization for mitigating hallucinations in multimodal large language models." Proceedings of the AAAI Conference on Artificial Intelligence. Vol. 39. No. 6. 2025.
- Liu, Jiazhen, et al. "PhD: A ChatGPT-Prompted Visual Hallucination Evaluation Dataset." Proceedings of the Computer Vision and Pattern Recognition Conference. 2025.

**Questions:**

* Have the authors analyzed the latency overhead introduced by using the T2I model during decoding? A discussion or measurement of this additional computational cost would be valuable.

---

> ### Author Response · Authors · 2025-11-20
>
> - **Q1：How does the proposed method differ from ConVis? Could the authors clarify the novelty of their approach in comparison?**
> - **A1**: Thanks for suggesting additional reference and comparisons.
>
> **1.** Although both methods leverage a T2I model to mitigate hallucination in VLMs, the two works fundamentally target different subfields and operate at different stages of the generation pipeline.
> - **Conceptual difference**: Our method proposes a steering-based mechanism that identifies a “steering vector” capable of modulating the model’s internal representation before the decoding stage. The goal is to adjust the model’s behavior by injecting a direction into its hidden space. In contrast, ConVis is a contrastive decoding method. It modifies the output decoding process and does not attempt to change or steer the model’s internal representations.
> - **Operational difference**: our method is lightweight and works with only one generated image. ConVis requires the simultaneous generation of four images, which substantially increases computation and latency, making it less practical for deployment. Thus, our approach introduces less overhead while achieving strong mitigation of hallucinations.
> - **Novelty**: To the best of our knowledge, we are the first to introduce a steering-vector–based approach for hallucination correction in VLMs using a T2I model. Prior work, including ConVis, has only explored decoding strategies rather than steering internal representations.
>
> **2.**: To fully address your comment, we compare our work with ConVis using the POPE (MSCOCO) and MME benchmark under the same experimental settings (using LLaVA-v1.5-7B) in terms of both performance and overhead. As shown in the table below, our proposed method suppresses ConVis by a large margin.
>
> **Table A-1: Comparison of ConVis on POPE and MME on LLaVA-v1.5-7B**
> | | **POPE** | | **MME** | | | | |
> |:---|---:|---:|---:|---:|---:|---:|---:|
> | | Accuracy | F1-score | Existence | Count | Position | Color | Total |
> | baseline | 79.83 | 79.29 | 190.0 | 153.3 | 133.3 | 155.0 | 643.3 |
> | ConVis | - | 83.00 | **195.0** | **158.3** | 133.3 | 155.0 | 653.6 |
> | Ours | **85.70** | **86.14** | 193.0 | 156.7 | **150.0** | **167.7** | **667.3** |
>
> **Table A-2: Comparison of decoding cost with ConVis per instance (seconds per instance)**
> | | POPE | MME |
> |:---|---:|---:|
> | ConVis | 9.792s | 9.951s |
> | Ours | **8.073s** | **8.115s** |
>
> **3.** We have included the suggested work in Section 2 of the revised manuscript. Specifically,
>   - Sec 2, paragraph 3, line 152-154: add reference *“Park, Yeji, et al. "Convis: Contrastive decoding with hallucination visualization for mitigating hallucinations in multimodal large language models." Proceedings of the AAAI Conference on Artificial Intelligence. Vol. 39. No. 6. 2025.”*
> - **Q2: How does the dependence on the T2I model’s quality affect the robustness of the proposed method, and would using multiple generated images to average results improve reliablity?**
> - **A2:** Thanks for the question. In addition to the results using another T2I model GPT-image, in Table A-3, we use the same T2I model as the suggested work ConVis [a], which is Hyper-SDXL and another outdated model DALL-E3. It can be seen that our method maintains high stability across different T2I models. Moreover, to fully address your comments, we evaluate the performance using mutiple generated images to average results in Tavle A-4. It can be seen that the numbers under multiple repeated experiments are quite close to the numbers in the paper under default settings.
>
> **Table A-3: Stability analysis across different T2I models*8**
> | | Baseline | FLUX.1_dev | GPT-image | Hyper-SDXL | DALL-E3
> |:---|---:|---:|---:|---:|---:|
> | $CHAIR_s$ | 53.0 | 47.8 | 48.6 | 46.2 |49.9|
> | $CHAIR_i$ | 14.0 | 12.6 | 13.2 | 11.9 |12.8|
> | Recall | 81.0 | 80.9 | 78.7 | 79.7 |78.8|
>
> **Table A-4: Performance of using multiple generated images to average.**
>
> | | $CHAIR_S$ | $CHAIR_I$ | Recall |
> |:---|:---:|:---:|:---:|
> | baseline | 53.0 | 14.0 | 81.0 |
> | Ours | 47.8 ± 0.23 | 12.7 ± 0.19 | 80.10 ± 0.09 |

---

> ### Author Response · Authors · 2025-11-20
>
> - **Q3: Permanence on Other Benchmarks.**
> - **A3:** Thank you for your advice. We evaluate the performance of our work on three other benchmarks, i.e., GAVIE , MMBench, and Faithscore, using LLaVA-v1.5-7B.
> As shown in Table A-5/6, it can be seen that our method achieves superior effectiveness of  hallucination mitigation across different benchmarks. **We have included these results in Appendix E in the revised manuscript.**
>
> **Table A-5: Performance of our method using GAVIE and FaithScore.**
>
> | | GAVIE | | FaithScore | |
> |:---|:---:|:---:|:---:|:---:|
> | | Accuracy | Relevancy | Faithscore | Faithscore in sentence |
> | Baseline | 4.360 | 6.110 | 0.88 | 0.62 |
> | Ours | **5.635** | **6.215** | **0.89** | **0.70** |
>
> **Table A-6: Performance of our method using MMBench.**
>
> | | Overall | CP | FP-S | FP-C | AR | LR | RR |
> |:---|:---:|:---:|:---:|:---:|:---:|:---:|:---:|
> | Ours | **58.97** | **69.59** | **60.05** | **51.42** | **74.31** | **31.21** | 44.08 |
> | Baseline | 50.95 | 53.32 | 56.28 | 47.37 | 60.76 | 21.39 | **50.71** |
>
> - **Q4: Inconsistent terms of LVLM and MLLM.**
> - **A4:** Thank you for pointing this out. We have revised this in the revised manuscript.
> - **Q5: Analysis on the latency overhead of using T2I models.**
> - **A5:** Thanks for the question. We agree that incorporating a T2I model introduces additional latency during inference.  However, the T2I model is only used to derive the steering vector, and the steering operation itself is lightweight at inference. The T2I invocation is therefore a one-time cost, not a per-token decoding cost, making the overall overhead practical for real-world use. **We have expanded our discussion to discuss this point.**

---

> ### Author Response · Authors · 2025-11-23
>
> Dear reviewer jLcd:
>
> Thanks so much again for the time and effort in our work. According to the comments and concerns, we conduct the corresponding experiments and further discuss the related points. Additionally, according to your suggestions, we have revised the manuscript and added more experiments in Appendix E.
>
> As the discussion period is about to close, may I know if our rebuttal addresses the concerns? If there are further concerns or questions, we are willing to address them. Thanks again for taking the time to review our work and provide insightful comments.

---

> > ### Comment · Reviewer_jLcd · 2025-11-24
> > **Thank you for your response**
> >
> > Thank you for providing a response to my earlier comments.
> >
> > Nevertheless, even after reading your rebuttal, I still find the novelty of the proposed research to be quite limited. Below are the two contribution statements taken directly from your submission.
> >
> > ----------------
> >
> > 2) We introduce a novel hallucination amplification mechanism that projects caption semantics into the visual space using a T2I model. This makes otherwise implicit hallucinations perceptible and provides a lightweight way to construct reliable supervisory signals.
> > 3) Our method jointly anchors semantics from the original images and suppresses hallucination directions from the reconstructed image. This dual guidance removes only hallucination components while preserving genuine semantics, striking a balance between faithfulness and informativeness.
> >
> > ----------------
> >
> > In my view, both of these are largely inherited from ConVis, and the distinction you highlighted in the rebuttal—namely that your method performs steering at the vector level using the reconstructed image—suggests that the originally stated contributions are in fact narrower than implied. Moreover, the idea of deriving a supervised signal from reconstructed images has already been explored in ESREAL (ECCV’24). For these reasons, I believe the manuscript requires substantial revision to more clearly articulate the genuine novelty and contribution of your work.
> >
> >
> >
> > Given the current overlap with prior work and the limited differentiation presented, I strongly encourage a thorough clarification and reframing of the contributions.

---

> ### Author Response · Authors · 2025-11-29
>
> Thanks for the comment.
>
> It seems like the most important concern is the lack of novelty.
>
> Although ConVis and ESREAL both leverage T2I models for semantic reconstruction, this overlap does **not in any way downplay** the contribution or novelty of our approach. Our method is the **FIRST** work to empower steering techniques with semantic reconstruction. We propose a novel way to create contrastive signal with **Generative Anchor** during steering, which is **unsupervised and powerful**. ConVis, ESREAL, and our method belong to **entirely different research sub-domains**. What we introduce is the **first complete application and practical implementation** of semantic reconstruction for steering, something that **ConVis and ESREAL do not address at all**.

---

### Author Response · Authors · 2025-12-03

# General Description
Dear Area Chair:

We sincerely thank all Reviewers and Area Chair for the time and effort dedicated to reviewing our paper. We appreciate the constructive feedback from all reviewers --- jLcd, vnXo, xReZ, and bvix, which has helped us further improve the clarity and presentation of the manuscript. We also thank the reviewers for acknowledging the strengths of our work, including: **1. Efficient and scalable solution**(R1, R2, R3)**, 2. a novel T2I-driven contrastive steering method**(R2, R3), **3. Effective and fine-grained hallucination mitigation**(R2) and **4. Extensive experiments and thorough analysis, and well-written paper**(R4).

The concerns are mainly concentrated on more comparisons with prior works and generalization analyses across benchmarks.

## Additional Experiments and Analyses:

In our responses, we did additional experimental results and analyses that address the comments, including:
1. Performance on more extensive benchmarks(MMBench, GAVIE, Faithscore) and consistently outperform baseline well.(R1, R2, R3)
2. Methodological and empirical comparison with ConVis(another work leverage T2I model) on efficiency and accuracy.(R1)
3. Performance results using diverse T2I models.(R1)
4. Comparison with training-based methods(PertuboLLaVA and RLAIF-V) and training-free method(VTI) and consistently outperform others well.(R4)

**All corresponding results have been incorporated into the revised manuscript.** Moreover, we have also included more discussions, clarified method details, and provided suggestions for parameter settings. These revisions address all reviewer comments in detail.

Thank you again for your insightful feedback and for helping us further refine our work.

Sincerely,

Authors of Submission 11992

---

### Meta-Review · Area_Chair_wbaL · 2026-01-06

**Summary:**

The paper proposes a training-free contrastive decoding method that leverages a text-to-image (T2I) model to mitigate hallucinations in large vision-language models (LVLMs). The approach reduces object-, attribute-, and relation-level hallucinations without finetuning or performance loss.


The reviewers present concerns on limited novelty with highly relevant prior work [reviewer jLcd].
The proposed method is not efficeint, scalable, and general [Reviewer bvix]
The performance relies heavily on T2I models. [reviewer jLcd, reviewer xReZ]
The evaluation are not sufficient, missing some benchmarks and recent mainstream methods. [Reviewer jLcd, vnXo, XReZ, bvix]

Considering the reviewers comments as well the authors rebuttal information, I have the same feeling that the technical novelties are limited, specifcally comparing with ConVis. And the experiment settings and improvements are limited. As such, I am considering to reject the paper.

**Reviewer Concerns:**

The limitations on the technical novelities are still outstanding.

And evaluation can be somewhat addressed.

**Reviewer Scores:**

I think the reviewers with negative ratings will not change their score, as they express their main concerns on the technical novelties.

---

### Decision · Program_Chairs · 2026-01-26

Reject